# Compound Phenotype Due to Recessive Variants in *LARP7* and *OTOG* Genes Disclosed by an Integrated Approach of SNP-Array and Whole Exome Sequencing

**DOI:** 10.3390/genes11040379

**Published:** 2020-03-31

**Authors:** Pietro Palumbo, Orazio Palumbo, Maria Pia Leone, Ester di Muro, Stefano Castellana, Luigi Bisceglia, Tommaso Mazza, Massimo Carella, Marco Castori

**Affiliations:** 1Division of Medical Genetics, Fondazione IRCCS-Casa Sollievo della Sofferenza, 71013 San Giovanni Rotondo (Foggia), Italy; p.palumbo@operapadrepio.it (P.P.); mariapia_leone@hotmail.it (M.P.L.); dimuro.ester@gmail.com (E.d.M.); l.bisceglia@operapadrepio.it (L.B.); m.carella@operapadrepio.it (M.C.); m.castori@operapadrepio.it (M.C.); 2Bioinformatics Unit, Fondazione IRCCS-Casa Sollievo della Sofferenza, 71013 San Giovanni Rotondo (Foggia), Italy; s.castellana@css-mendel.it (S.C.); t.mazza@operapadrepio.it (T.M.)

**Keywords:** Alazami syndrome, compound phenotype, *OTOG*, regions of homozygosity, whole-exome sequencing

## Abstract

Neurodevelopmental disorders are a challenge in medical genetics due to genetic heterogeneity and complex genotype-phenotype correlations. For this reason, the resolution of single cases not belonging to well-defined syndromes often requires an integrated approach of multiple whole-genome technologies. Such an approach has also unexpectedly revealed a complex molecular basis in an increasing number of patients, for whom the original suspect of a pleiotropic syndrome has been resolved as the summation effect of multiple genes. We describe a 10-year-old boy, the third son of first-cousin parents, with global developmental delay, facial dysmorphism, and bilateral deafness. SNP-array analysis revealed regions of homozygosity (ROHs) in multiple chromosome regions. Whole-exome sequencing prioritized on gene-mapping into the ROHs showed homozygosity for the likely pathogenic c.1097_1098delAG p. (Arg366Thrfs*2) frameshift substitution in *LARP7* and the likely pathogenic c.5743C>T p.(Arg1915*) nonsense variant in *OTOG*. Recessive variants in *LARP7* cause Alazami syndrome, while variants in *OTOG* cause an extremely rare autosomal recessive form of neurosensorial deafness. Previously unreported features were acrocyanosis and palmoplantar hyperhidrosis. This case highlights the utility of encouraging technological updates in medical genetics laboratories involved in the study of neurodevelopmental disorders and integrating laboratory outputs with the competencies of next-generation clinicians.

## 1. Introduction

Neurodevelopmental disorders (NDDs) are the most frequent cause of disability in children and, currently, the main reason for referral in clinical genetic services. Nevertheless, they remain a major challenge due to high genetic-heterogeneity and weak genotype-phenotype correlations in most cases. The introduction of next-generation technologies has been demonstrated as effective in reducing the costs of the diagnostic trajectory of intellectual disability and in clinical decision-making for focal epilepsy [1,2]. In addition, whole-exome sequencing (WES)-based diagnostics is predicted to increase the number of individuals in whom the suspected pleiotropic syndrome is resolved as compound phenotypes, due to the simultaneous involvement of two or more disease-genes with separated biological functions (i.e., compound phenotypes). In this context, consanguinity facilitates the concurrence of multiple autosomal recessive disorders in the same individual [3]. 

Alazami syndrome (MIM 615071) is an ultra-rare autosomal recessive syndromal form of NDD, described for the first time in 2012 in an inbred Saudi Arabian family with a novel form of primordial dwarfism, due to a homozygous variant in *LARP7* (OMIM 612026) gene [4]. Twenty-four patients have been reported to date [5], often presenting subtle but recurrent facial features, such as deep-set eyes, broad nose, wide mouth and teeth anomalies in addition to short stature, intellectual disabilities/global developmental delay and poor speech. *LARP7* encodes for the La-related protein 7, which acts as a chaperone of the noncoding RNA 7SK [4]. Alazami syndrome is usually caused by frameshift variants in *LARP7* [5]. Neurosensorial deafness is not a feature of Alazami syndrome.

*OTOG* gene (OMIM 604487) encodes for the Otogelin protein, a noncollagenous component of the acellular structures that cover the sensory epithelia of the inner ear [6]. Recessive variants in *OTOG* are a very rare cause of hereditary hearing loss, known as DFNB18B (MIM 614945) [7]. To date, only four causative variants in *OTOG* have been found in seven patients with DFNB18B: two nonsense, one frameshift, and one missense variant [8]. 

Here, we report a 10 year-old boy, born to consanguineous parents, with global developmental delay, absent speech, facial dysmorphisms, and bilateral congenital hearing loss. Multiple regions of homozygosity (ROH) identified by SNP-array-supported whole-exome sequencing analysis, allowing us to identify two homozygous deleterious variants in *LARP7* and *OTOG*. This study reinforces the utility of combining high-resolution SNP-array technologies with WES in resolving highly complex phenotypes, especially in the presence of parental consanguinity. 

## 2. Materials and Methods 

### 2.1. Genomic DNA Extraction and Quantification

This family provided signed informed consent to molecular testing and to the full content of this publication. This study was conducted in accordance with the 1984 Declaration of Helsinki and its subsequent revisions. Molecular testing carried out in this report is based on the routine clinical care of our institution. Peripheral blood samples were taken from both the patient and his parents, and genomic DNA was isolated by using Bio Robot EZ1 (Quiagen, Solna, Sweden). The quality of DNA was tested on 1% electrophorese agarose gel, and the concentration was quantified by Nanodrop 2000 C spectrophotometer (Thermo Fisher Scientific, Waltham, MA, USA).

### 2.2. SNP Array Analysis 

SNP array analysis of the proband and his parents was carried out with the CytoScan HD Array (Thermo Fisher Scientific) as previously described [9]. Data analysis was performed using Chromosome Analysis Suite Software version 4.0 (Thermo Fisher Scientific) following a standardized pipeline. Briefly: (i) the raw data file (.CEL) was normalized using the default options; (ii) an unpaired analysis was performed using 270 HapMap samples as a baseline in order to obtain copy numbers value and regions of homozygosity (ROHs) from .CEL files. The amplified and/or deleted regions were detected using a standard Hidden Markov Model (HMM) method. Size threshold for analysis was kept as 5 Kb for copy number variations (CNVs), and 1 Mb for ROHs. In order to identify clinical or functionally relevant genomic variants, we compared all chromosomal alterations identified to those collected in our internal database of ~4,000 patients studied by SNP Arrays since 2010 and public databases, including the Database of Genomic Variants (DGV; available online at: http://projects.tcag.ca/variation/), DECIPHER (available online at: https://decipher.sanger.ac.uk/) and ClinVar (available online at: https://www.ncbi.nlm.nih.gov/clinvar/). Base pair positions, information about genomic regions and genes affected by CNVs and/or ROHs, and known associated diseases will be derived from the University of California Santa Cruz (UCSC) Genome Browser (available online at: http://genome.ucsc.edu/cgi-bin/hgGateway), build GRCh37 (hg19). The clinical significance of each rearrangement detected was assessed following the American College of Medical Genetics (ACMG) guidelines [10].

### 2.3. Whole-Exome Sequencing 

Proband DNA was analyzed by WES by using SureSelect Human Clinical Research Exome V6 (Agilent Technologies, Santa Clara, CA, USA) following manufacturer instructions. This is a combined shearing-free transposase-based library prep and target-enrichment solution, which enables comprehensive coverage of the entire exome. This system enables a specific mapping of reads to target deep coverage of protein-coding regions from RefSeq, GENCODE, CCDS, and UCSC Known Genes, with excellent overall exonic coverage and increased coverage of HGMD, OMIM, ClinVar, and ACMG targets. Sequencing was performed on a NextSeq 500 System (Illumina, San Diego, CA, USA) by using the High Output flow cells (300 cycles), with a minimum expected coverage depth of 70x. All variants obtained from WES were called by means of the HaplotypeCaller tool of GATK ver. 3.58 [11] and were annotated based on frequency, impact on the encoded protein, conservation, and expression using distinct tools, as appropriate (ANNOVAR, dbSNP, 1000 Genomes, EVS, ExAC, ESP, KAVIAR, and ClinVar) [12,13,14,15,16], and retrieving pre-computed pathogenicity predictions with dbNSFP v 3.0 (PolyPhen-2, SIFT, MutationAssessor, FATHMM, LRT and CADD) [17] and evolutionary conservation measures. 

Next, variants prioritization was performed starting from homozygous variants, involving the genes located in the ROHs identified. Firstly, variants described as benign and likely benign were excluded. Then, remaining variants were classified based on their clinical relevance as pathogenic, likely pathogenic, or variant of uncertain significance according to following criteria: (i) nonsense/frameshift variant in genes previously described as disease-causing by haploinsufficiency or loss-of-function; (ii) missense variant located in a critical or functional domain; (iii) variant affecting canonical splicing sites (i.e., ±1 or ±2 positions); (iv) variant absent in allele frequency population databases; (v) variant reported in allele frequency population databases, but with a minor allele frequency (MAF) significantly lower than expected for the disease (<0.002 for autosomal recessive disease and <0.00001 for autosomal dominant disease); (vi) variant predicted and/or annotated as pathogenic/deleterious in ClinVar and/or LOVD; (vii) variants in *GJB2* gene setting up as cut off a MAF <= 0.1 and variants for other genes associated with recessive hearing impairment and reported in Hereditary Hearing Loss Database (https://hereditaryhearingloss.org/) setting up as cut off a MAF <= 0.5. 

The resulting putative pathogenic variants were confirmed by Sanger sequencing in both the proband and the parents’ DNA. PCR products were sequenced by using BigDye Terminator v1.1 Sequencing Kit (Applied Biosystems, Foster City, CA, USA) and ABI Prism 3100 Genetic Analyzer (Thermo Fisher Scientific). The clinical significance of the identified putative variants was interpreted according to the American College of Medical Genetics and Genomics (ACMG) [18]. Variant analysis was carried out considering the ethnicity of the patient.

### 2.4. Variant Designation

Nucleotide variants nomenclature follows the format indicated in the Human Genome Variation Society (HGVS, http://www.hgvs.org) recommendations and reported in the Leiden Open Variation Databases (LOVD) (https://databases.lovd.nl/shared/individuals/00276129).

## 3. Results

### 3.1. Clinical Description

The patient is a 10 year-old male, son of first cousins, unaffected parents of Caucasian origin (Southern Italy). His family history was unremarkable. The patient was born at term (39 weeks) with a low birth weight (2330 g). His birth length, head circumference, and Apgar score were unavailable. The neonatal period ran physiologically well, except for patent foramen ovale, but the following early psychomotor development was delayed. The patent foramen ovale resolved spontaneously in the following months. The patient held his head at 7 months, sat alone at about 18 months, and walked at 24 months. He never attained fully autonomous walking, language skills, or sphincter control. Due to the severe speech impairment, the patient underwent multiple audiological exams which diagnosed a profound bilateral neurosensorial deafness. Although the patient received auditory prosthesis, their language skills did not improve. At 1 year, a brain MRI scan was performed and did not reveal abnormalities except for a mild myelination delay. At examination, his height was 98 cm (<3rd centile), weight 19.8 kg (<3rd centile) and his head circumference was 49.5 cm (<3rd centile). His facial features included sparse eyebrows, prominent supraorbital ridges, short nose, high palate, and full lips. His language was absent. His gait was unstable with a broad base and mild spasticity of the lower limbs. The patient also displayed acrocyanosis and marked palmoplantar hyperhidrosis. The parents said that these two features were present since his infancy.

### 3.2. Molecular Findings

We did not identify any clinically significant CNV in the patient, while multiple regions of ROH were detected on chromosomes 2, 3, 4, 7, 8, 11, 16 and 18, respectively. The regions of homozygosity identified in the patient are depicted in Figure 1 and listed in Table 1. 

This finding was made in accordance with parental consanguinity and offered us the opportunity to prioritize variant filtering from the WES data for homozygosity in disease-gene-mapping within these regions. Accordingly, WES allowed us to detect two previously unpublished, homozygous variants: a frameshift variant c.1097_1098del, p.(Arg366Thrfs*2) in *LARP7* (NM_015454.2) mapping in the ROH region on chromosome 4, and a nonsense variant c.5743C>T, p.(Arg1915*) in *OTOG* (NM_001277269.1) mapping to the ROH region on chromosome 11. No further clinically relevant variant was identified at the homozygous state in the other ROH regions. Variants mapped outside these regions were prioritized according to the criteria listed in the Methods sections. No variant was identified as reaching the likely pathogenic or pathogenic rank according to the American College of Medical Genetics and Genomics and compatible with the observed phenotype. 

The identified variants in *LARP7* and *OTOG* were confirmed by Sanger sequencing (Figure 2) using specific primers (*LARP7*, exon 8, Forward Primer: GAATCCCTAGCTCCCCGATC; *LARP7,* exon 8, Reverse Primer: GTGCAGTTCTTGGCTACAGG. ***OTOG****,* exon 35, Forward Primer: CACTTAGCCCAGTACTGCCT; ***OTOG****,* exon 35, Reverse Primer: CTCAGGGCATAGGATGTGGG).

The *LARP7* variant c.1097_1098del, p.(Arg366Thrfs*2) had an MAF of 0.00002 and 0.00003 in gnomAD and ExAC, respectively. The *OTOG* variant c.5743C>T, p.(Arg1915*) had an MAF of 0.00003 and 0.0001 in gnomAD and ExAC, respectively. The *LARP7* variant is predicted to cause a frameshift leading to the deletion of the last 217 amino acids of the protein. The *OTOG* variant is predicted to truncate the protein and to cause loss of the last 1011 amino acids. Both variants were classified as likely pathogenic according to ACMG guidelines [18]. Complete bioinformatic details of these variants are reported in Table 2. No further variant classified as pathogenic or likely pathogenic, according to ACMG guidelines in other genes and previously associated with phenotypes compatible with the clinical features reported by the patients, were identified by the bioinformatics analysis. In particular, we did not find any additional deleterious variants in other genes responsible for hereditary hearing loss and annotated in the Hereditary Hearing Loss Database (https://hereditaryhearingloss.org/), or the other satellite symptoms, including palmoplantar hyperhidrosis and acrocyanosis.

## 4. Discussion

Here, we report a boy with severe NDD, absent speech with profound neurosensorial deafness, growth restriction, and facial dysmorphisms, resulting from double homozygosity in *LARP7*, causing Alazami syndrome and *OTOG*, associated with a very rare form of autosomal recessive neurosensorial deafness. Although the parental consanguinity predicted multiple ROHs at the SNP array analysis, it did not predict that there were multiple genes involved in the disease. Given our experience, we suggest, in cases like the one we describe, that the possibility of multiple genes involved in the etiology of the phenotype should be taken into account. 

*LARP7* gene encodes a chaperone protein required for both stability and function of the RNA. In particular, it forms a complex with the nuclear 7SK RNA to regulate RNA polymerase II transcription. To date, the clinical effect of deleterious variants in *LARP7* is restricted to Alazami syndrome. Currently, 24 patients from 12 families are described as having Alazami syndrome and recessive variants in *LARP7* [5]. Of the 14 identified variants, 12 were frameshift, one was nonsense, and one was predicted to cause abnormal splicing. Therefore, all deleterious variants causing Alazami syndrome are presumably null alleles [5]. Our findings are in line with the current literature, as the variant detected in our patient is a frameshift predicted to cause the deletion of the last 217 amino acids and, therefore, the loss of the RNA recognition motif 2 (RRM2). At a functional level, this event would likely lead to the formation of a truncated protein [p.(Arg366Thrfs*2)] or nonsense-mediated mRNA decay. This previously unpublished variant is located near to the variants reported in the original report, as well as those described by Hollink and colleagues [19]. This brings the number of identified deleterious variants in *LARP7* to 15 (Figure 3). 

The La protein RNA-binding motif is critical for the physiological function of the encoded protein as all identified deleterious variants cause the loss of this domain. On a clinical perspective, the overall phenotype observed in this patient is fully in agreement with the Alazami syndrome clinical spectrum, which mainly includes global developmental delay with absent speech, growth retardation (of prenatal onset), and peculiar facial appearance. Interestingly, the described patient also presented with bilateral profound neurosensorial deafness, acrocyanosis, and palmoplantar hyperhidrosis. While hearing impairment may be explained by the combined recessive variants in *OTOG* (see below), acrocyanosis and palmoplantar hyperhidrosis remain unexplained. The concurrence of these features might be causal, as both are unspecific and are common in the general population. Nevertheless, at the moment, we cannot exclude that acrocyanosis and palmoplantar hyperhidrosis might represent unusual phenotypic manifestations of *LARP7*, *OTOG,* or a combination of both. Further patients are needed in order to explore whether they are casually associated with Alazami syndrome or, rather, represent its unusual manifestations. 

*OTOG* gene encodes for the Otogelin, an N-glycosylated protein that is expressed in the acellular membranes covering the six sensory epithelial patches of the inner ear, including the cochlea, the tectorial membrane over the organ of Corti, the vestibule, the otoconial membranes over the utricular and saccular maculae, and the cupulae over the cristae ampullares of the three semicircular canals. These membranes are collectively involved in the mechanotransduction process of external auditory stimuli. The movement of these membranes, which is induced by sound in the cochlea or acceleration in the vestibule, results in the deflection of the stereocilia bundle at the apex of the sensory hair cells, which in turn opens the mechanotransduction channels located at the tip of the stereocilia [6]. Several studies aimed at evaluating the expression level of otogelin in human tissues found that this protein is ubiquitous, but the transcript levels were the highest in the inner ear, followed by the kidney, the lung, the spleen, the thymus, and the liver [7]. To date, deleterious variants in *OTOG* are selectively associated with a rare non-syndromal form of autosomal recessive neurosensorial deafness. The present patient and previously published deleterious variants bring the number of described patients to eight and the number of reported variants to five (Figure 4). 

The c.5743C>T variant is predicted to introduce a premature stop codon, which would, presumably, lead to the formation of a truncated protein [p.(Arg1915*)] or nonsense-mediated mRNA decay. This is in line with the predominance of null alleles in *OTOG* associated with hereditary deafness. In our patient, the hearing phenotype is apparently severe and of congenital onset, as suggested by previous papers speculating on the prenatal onset of the disease in *OTOG*-related autosomal recessive hereditary deafness [7,8]. 

## 5. Conclusions

Our case highlights the importance of combined genomic approaches in solving complex phenotypes and the role of parental consanguinity in prioritizing candidate gene-mapping in ROHs. We also identified novel deleterious variants in *LARP7* and *OTOG*, which expand their molecular spectrum and confirm the associated phenotypes. This patient also presented acrocyanosis and palmoplantar hyperhidrosis, which might represent unusual phenotypic manifestations of *LARP7*, *OTOG,* or a combination of both. 

## Figures and Tables

**Figure 1 genes-11-00379-f001:**
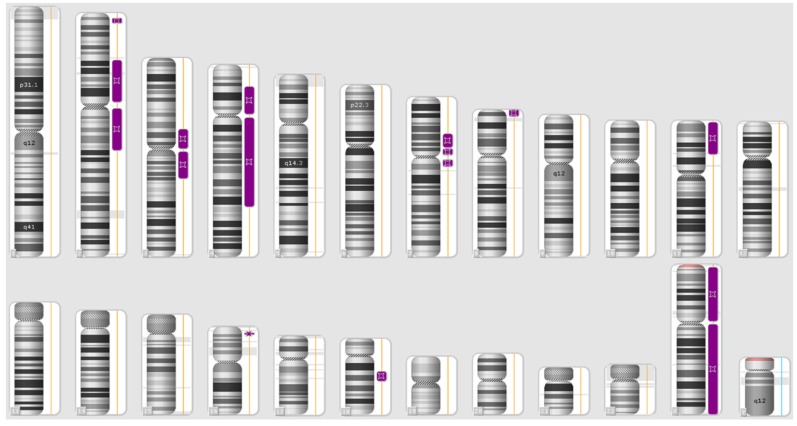
Karyoview of the patient. Regions of homozygosity (ROHs) identified in the proband genome are indicated by purple bars.

**Figure 2 genes-11-00379-f002:**
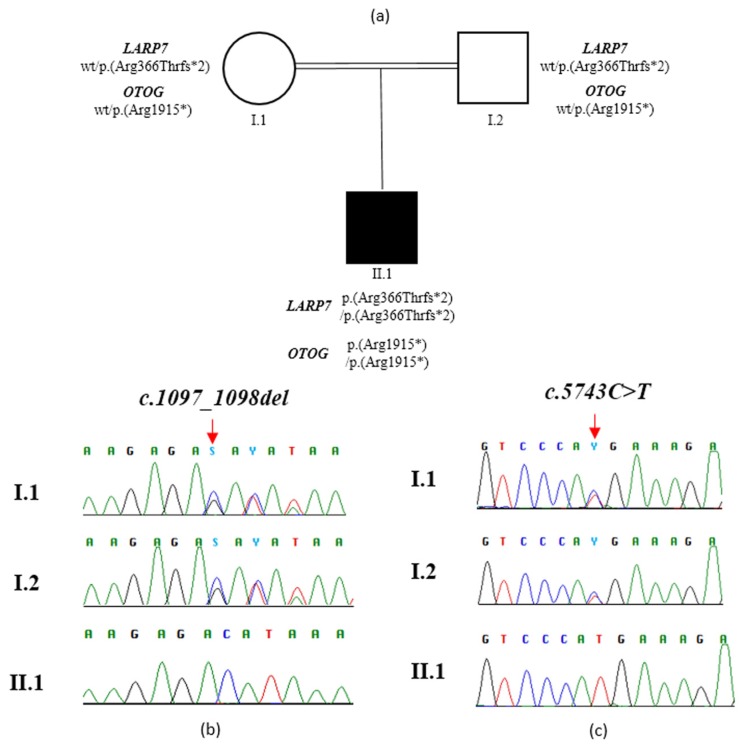
Family tree and molecular findings in the family. (**a**) Double lines in pedigree indicate consanguinity. Filled and unfilled circles/squares represent affected and unaffected individuals respectively. (**b**) Sanger sequence of a PCR product amplified with primers targeting exon 8 of *LARP7* of individual II.1 and his unaffected parents. (**c**) Sanger sequence of a PCR product amplified with primers targeting exon 35 of *OTOG* of individual II.1 and his unaffected parents. Vertical red arrows indicate variant position.

**Figure 3 genes-11-00379-f003:**
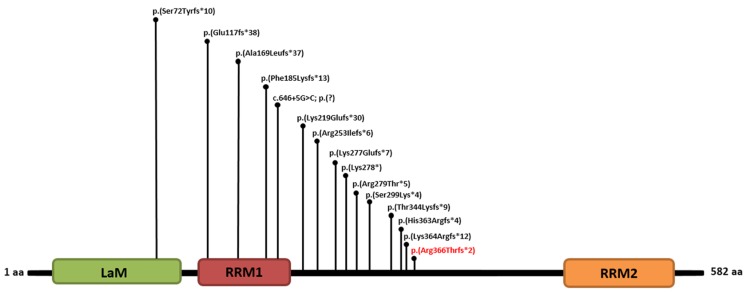
Schematic representation of the protein domains of La-related protein 7 [20,21], encoded by *LARP7*, with positions of all known deleterious variants published to date and including the present one [4,5,19,22,23,24,25,26]. The variant identified here is in red. (LaM, Lupus antigen Motif; RRM1, RNA Recognition Motif 1; RRM2, RNA Recognition Motif 2).

**Figure 4 genes-11-00379-f004:**
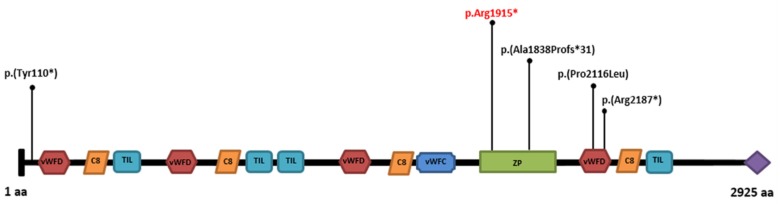
Schematic representation of protein domains of Otogelin [27], encoded by *OTOG*, with positions of all known deleterious variants published so far, and including the present one [7,8]. The variant identified here is in red. (vWFD, von Willebrand factor type D domain; C8, cystein-rich domain; TIL, trypsin inhibitor-like cysteine-rich domain; vWFC, von Willebrand factor type C domain; ZP, zona pellucida domain; C-terminal cysteine knot-like domain).

**Table 1 genes-11-00379-t001:** List of regions of homozygosity (ROHs) identified in the proband genome according to the International System for Human Cytogenetic Nomenclature (ISCN 2016). Size and number of genes included in each ROH are indicated. All genomic coordinates were based on the GRCh37/hg19 build of the Human Genome.

Regions of Homozigosity (ISCN 2016)	Size (Mb)	Number of Genes
*arr[GRCh37] 2p25.2p25.1(5573938_10537316)x2 hmz*	4.9	40
*arr[GRCh37] 2p21p11.2(47453937_89129064)x2 hmz*	41.6	317
*arr[GRCh37] 2q11.1q22.1(95341387_137166151)x2 hmz*	41.8	340
*arr[GRCh37] 3p13p11.1(71199337_90485635)x2 hmz*	19.2	49
*arr[GRCh37] 3q11.1q13.33(93536053_120421935)x2 hmz*	26.8	173
*arr[GRCh37] 4p15.2p11(21674784_49089181)x2 hmz*	27.4	117
*arr[GRCh37] 4q11q31.1(52686799_141024195)x2 hmz*	88.3	468
*arr[GRCh37] 7p14.2p12.2(36814859_50418506)x2 hmz*	13.6	98
*arr[GRCh37] 7p12.1p11.1(51736013_58019983)x2 hmz*	6.2	38
*arr[GRCh37] 7q11.21q11.22(62461703_68908285)x2 hmz*	6.4	50
*arr[GRCh37] 8p23.3p23.1(168483_7011075)x2 hmz*	6.8	46
*arr[GRCh37] 11p15.5p13(1751363_33420180)x2 hmz*	31.6	367
*arr[GRCh37] 16p13.3p13.2(6643315_8047081)x2 hmz*	1.4	1
*arr[GRCh37] 18q12.2q21.1(34083335_43959703)x2 hmz*	9.8	33

Mb, Megabases.

**Table 2 genes-11-00379-t002:** Characteristics of the variants identified in the *LARP7* and *OTOG* genes.

Chromosome	Start	End	Reference Allele	Alternative Allele	Genotype	Gene	Nucleotide Change	Amino Acid Change	Gene Impact	dbSNP ID	gnomAD_exome Allele Count	TOPMED Allele Count	ExAC_ALL Allele Count
4	113568939	113568941	AAG	A	homo_alt	*LARP7* *NM_015454.2*	c.1097_1098del	p.(Arg366Thrfs*2)	frameshift substitution	rs566464249	4/237240 MAF 0.00002	6/125568 MAF 0.00005	3/113350 0.00003
11	17632554	17632554	C	T	homo_alt	*OTOG* NM_001277269.1	c.5743C>T	p.(Arg1915*)	stopgain	rs761287044	5/146398 MAF 0.00003	2/125568 MAF 0.00002	1/15676 0.0001

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
