# Peer review of "Compound Phenotype Due to Recessive Variants in LARP7 and OTOG Genes Disclosed by an Integrated Approach of SNP-Array and Whole Exome Sequencing"

_genes, 2020, doi:10.3390/genes11040379_

Round 1
Reviewer 1 Report
The authors describe the association of defects in two genes with the symptoms in a child with a neurodevelopmental disorder. Homozygous truncating variants were identified in both LARP7 and OTOG. These are likely to explain the symptoms described for Alazami syndrome and hearing impairment. Acrocyanosis and palmoplantar hyperhidrosis have so far not been associated with defects in LARP7 or OTOG. Also, hearing impairment is more severe than so far described for defects of OTOG. Phenotyping and genotyping of individuals with rare syndromes is of high importance for counseling and medical care of patients and to get insight into symptoms that can be associated with a syndrome/disease and which symptoms of patient are likely to have a different cause.
Although the study has been well performed, a number of points need to be addressed.
Major:
It needs to be discussed/addressed whether acrocyanosis and/or palmplantar hyperhidrosis might be explained by homozygous or compound heterozygous (mild) variants in other genes. These symptoms are independently associated with other conditions. Can underlying defects in other genes be excluded? Did the authors look at compound heterozygous candidate variants?
The hearing impairment in the described patient is profound which is not the case in the so far described patients with defects in OTOG. Might variants in other genes known to be associated with hearing impairment contribute to the severity of this part of the phenotype? Homozygous or compound heterozygous variants in genes known to be associated with hearing impairment should be scrutinized. This is also true for variants in genes associated with dominant hearing impairment unless hearing in the parents was checked and found to be normal. Also, might defects in LARP7 contribute to the hearing phenotype?
The ethnicity of the patients should be indicated and taken into account for providing allele frequencies of the identified variants.
There are many mistakes in grammar and I will mention a few as examples
page 2: line59: 'this study reinforce' should be 'this study reinforces'.
page 2, line 66: 'the patient and their parents' should be 'the patient and his parents'.
page 2, line 77: 'deleted regions was' should be 'deleted regions were'
line 113: MRI was performed and resulted normal: replace 'resulted normal' by 'did not reveal abnormalities'
Minor
Page 3, line 91: which version of the SureSelect Human Clinical Research Exome kit was used?
line 95: remove the second 'targets'
line 113: which MAFs were used as cut off.
line 119/120 and throughout the text: American Colleague of Medical Genetics (ACMGG) should be American College of Medical Genetics (ACMG).
table 1 what do 'arr' and 'x2 hmz' mean? are these informative? otherwise remove.
Fig 2b: display a wt sequence instead of the 2nd sequence with the deletion. This would be more informative.
The first 10 lines of the discussion section repeat information of the introduction. This is redundant and can be largely removed.
Line 199: parental consanguinity predicts multiple ROHs but do not predict that there are multiple genes involved in a disease. It merely indicates that this should be considered. Please change the text.
lines 207-209: also NMD should be indicated which might lead to strongly reduced amounts of the protein.
Line 209: the second RNA-binding motif will be absent from the protein, RRM2? Terminology is inconsistent in text and figure/figure legend.
Legends figures 3 and 4: on which source is the domain structure of the proteins based?
Line 237: ref 20 does not contain the expression in the indicated tissues.
Author Response
To Editor-in-Chief
Genes: Special Issue "Selected Papers from the International Workshop on Fragile X and Other Neurodevelopmental Disorders"
March 16, 2020
Subject: genes-741715 revised version submission
Dear Editor,
Thank you very much for giving us an opportunity to submit the revised manuscript to your esteemed journal. In agreement with you, we did all the corrections suggested. The following are Point-by-point response to the Reviewer 1 criticism. We hope that these changes meet with your approval.
Author's Reply to the Review Report (Reviewer 1)
Although the study has been well performed, a number of points need to be addressed.
Major:
It needs to be discussed/addressed whether acrocyanosis and/or palmplantar hyperhidrosis might be explained by homozygous or compound heterozygous (mild) variants in other genes. These symptoms are independently associated with other conditions. Can underlying defects in other genes be excluded? Did the authors look at compound heterozygous candidate variants?
ANSWER: We did not find compound heterozygous or heterozygous variants in other genes after our variant evaluation pipeline. To explain this, we add the sentence: “No further variant classified as pathogenic or likely pathogenic according to ACMG guidelines in other genes previously associated with phenotypes compatible with the clinical features reported by the patients were identified by the bioinformatics analysis. In particular, we did not find any additional deleterious variants in other genes responsible for hereditary hearing loss or the other satellite symptoms, including palmoplantar hyperhidrosis and acrocyanosis.” (line 186-191).
The hearing impairment in the described patient is profound which is not the case in the so far described patients with defects in OTOG. Might variants in other genes known to be associated with hearing impairment contribute to the severity of this part of the phenotype? Homozygous or compound heterozygous variants in genes known to be associated with hearing impairment should be scrutinized. This is also true for variants in genes associated with dominant hearing impairment unless hearing in the parents was checked and found to be normal. Also, might defects in LARP7 contribute to the hearing phenotype?
ANSWER: We did not find compound heterozygous or heterozygous variants in other genes after our variant evaluation pipeline. To explain this, we add the sentence: “No further variant classified as pathogenic or likely pathogenic according to ACMG guidelines in other genes previously associated with phenotypes compatible with the clinical features reported by the patients were identified by the bioinformatics analysis. In particular, we did not find any additional deleterious variants in other genes responsible for hereditary hearing loss or the other satellite symptoms, including palmoplantar hyperhidrosis and acrocyanosis.” (line 186-191).
The ethnicity of the patients should be indicated and taken into account for providing allele frequencies of the identified variants.
ANSWER: We introduce this information on the first line of the section 3.1. Clinical Description: ” The patient is a 11-year-old male, son of fist cousins, unaffected parents Caucasian origin (Southern Italy)” and was taken into account for the allele frequency. Also, we added the sentence “Variant analysis was carried out considering the ethnicity of the patient” (line 122-123, page 3)
There are many mistakes in grammar and I will mention a few as examples
page 2: line59: 'this study reinforce' should be 'this study reinforces'. ANSWER: We changed the text as suggested
page 2, line 66: 'the patient and their parents' should be 'the patient and his parents'. ANSWER: We changed the text as suggested
page 2, line 77: 'deleted regions was' should be 'deleted regions were' ANSWER: We changed the text as suggested
line 113: MRI was performed and resulted normal: replace 'resulted normal' by 'did not reveal abnormalities' ANSWER: We changed the text as suggested
Minor
Page 3, line 91: which version of the SureSelect Human Clinical Research Exome kit was used? The version used was V6. ANSWER: We added this information in the section “2.3. Whole exome sequencing (WES)”, page 3, line 92
line 95: remove the second 'targets' ANSWER: We removed the word indicated
line 113: which MAFs were used as cut off. ANSWER: We used as cut off a MAF< 0.002 for autosomal recessive diseases and < 0.00001 for autosomal dominant disease. We added this information in the section “2.3. Whole exome sequencing (WES)”, page 3, lines 114-115 (< 0.002 for autosomal recessive disease and < 0.00001 for autosomal dominant disease).
line 119/120 and throughout the text: American Colleague of Medical Genetics (ACMGG) should be American College of Medical Genetics (ACMG). ANSWER: We changed the text as suggested. In particular we corrected the text at page 3, line 121-122, page 5, line 167, and page 6, line 185.
table 1 what do 'arr' and 'x2 hmz' mean? are these informative? otherwise remove. ANSWER: The Table 1 add information reporting in detail molecular data on the regions of homozygosity (ROH) identified by SNP-array. These findings were represented according with the International System for Human Cytogenetic Nomenclature (ISCN 2016) as indicated in the legend to Table 1. Following this System, arr = Microarrays, x2 hmz= homozygosity (or copy neutral loss of heterozygosity, i.e. loss of heterozygosity not due to copy number alterations).
Fig 2b: display a wt sequence instead of the 2nd sequence with the deletion. This would be more informative. ANSWER: We apologize for the error but we omitted to report in the image 2 the heterozygous state for the variant in OTOG detected in the mother (I.1) and the heterozygous state for the variant in LARP7 detected in the father (I.2). Taking into account these considerations, figures 2b and 2c are coherent with the written data while the figure 2a was not. Therefore, we added this information (i.e. the heterozygous state for both variants in both genes) in figure 2a for the subjects I.1 and I.2, respectively.
The first 10 lines of the discussion section repeat information of the introduction. This is redundant and can be largely removed. ANSWER: In agreement with the reviewer, we removed the first 10 lines of the discussion section (line195-204).
Line 199: parental consanguinity predicts multiple ROHs but do not predict that there are multiple genes involved in a disease. It merely indicates that this should be considered. Please change the text. ANSWER: Thank you very much for the suggestion. In agreement with the reviewer, we changed the sentence “This finding was predicted by the identification of multiple ROHs at the SNP array analysis and due to the known parental consanguinity” with the new one “Although the parental consanguinity predicted multiple ROHs at the SNP-array analysis, it did not predict that there were multiple genes involved in the disease. Given our experience, we suggest, in cases like the one we describe, that the possibility of multiple genes involved in the etiology of the phenotype should be take into account.”
lines 207-209: also NMD should be indicated which might lead to strongly reduced amounts of the protein. ANSWER: Given that both mutations generate a premature termination codon (PTC), and that in both cases the PTC originates more than 50 nucleotides from the splicing junction at 3 ', activation of the Nonsense-mediated mRNA decay (NMD) is likely [Isken O, Maquat LE. Quality control of eukaryotic mRNA: safeguarding cells from abnormal mRNA function. Genes Dev. 2007 Aug 1;21(15):1833-56]. In agreement with the reviewer, we re-write the paragraph from the line 219 to 223. The new one is: " Our findings are in line with the literature, as the variant detected in our patient is a frameshift predicted to lead to the deletion of the last 217 amino acids and, consequently, the RNA recognition motif 2 (RRM2), presumably leading to the formation of a truncated protein [p.(Arg366Thrfs*2)] or nonsense-mediated mRNA decay”. Moreover, at line 266-267 we state: “The c.5743C>T variant is predicted to introduce a premature stop codon presumably leading to the formation of a truncated protein [p.(Arg1915*)] or nonsense-mediated mRNA decay.”
Line 209: the second RNA-binding motif will be absent from the protein, RRM2? Terminology is inconsistent in text and figure/figure legend. ANSWER: Yes, the second RNA-binding motif will be absent from the protein. As suggested, we indicated this in the text adding this sentence: “Our findings are in line with the literature, as the variant detected in our patient is a frameshift predicted to lead to the deletion of the last 217 amino acids and, consequently, the RNA recognition motif 2 (RRM2)…..(line 219-221)” Also, we corrected the terminology in figure 3 and figure 3 legend.
Legends figures 3 and 4: on which source is the domain structure of the proteins based? ANSWER: In agreement with the editor, we changed the figures legend to include the sources used to create the domain structure of the proteins. In details, in Figure 3 we added two new references [20, 21] (also added in the reference list):
“20.Uchikawa, E., Natchiar, K.S., Han, X., Proux, F., Roblin, P., Zhang, E., Durand, A., Klaholz, B.P., Dock-Bregeon, A.C. Structural insight into the mechanism of stabilization of the 7SK small nuclear RNA by LARP7. Nucleic Acids Res. 2015, 43, 3373-3388.”
“21.Eichhorn, C.D., Chug, R., Feigon, J. hLARP7 C-terminal domain contains an xRRM that binds the 3' hairpin of 7SK RNA. Nucleic Acids Res. 2016, 44, 9977-9989.”
In Figure 4 we added the new reference [22] (also added in the reference list):
- Goodyear, R.J., Richardson, G.P. Structure, Function, and Development of the Tectorial Membrane: An Extracellular Matrix Essential for Hearing. Curr Top Dev Biol. 2018, 130, 217-244
Line 237: ref 20 does not contain the expression in the indicated tissues. ANSWER: We removed the incorrect reference 20 and indicate the correct one [7].

Reviewer 2 Report
The paper is thorough, interesting and well written.
The authors describe a case with novel deleterious variants in LARP7 and OTOG. This case underlines the importance of combined genomic approaches to solve complex phenotypes and the role of parental consanguinity in prioritizing candidate genes mapping in regions of homozygosity (ROHs). They point to the importance of SNP-Array in identifying these regions initially. This paper expands the molecular spectrum of the genes LARP7 and OTOG and confirm the associated phenotypes.
However, in order to improve the knowledge of these phenotypes it would be very useful to have the photographs of the patient. Furthermore, it would be helpful if the authors indicated the patient’s nationality.
The authors, moreover, state that nineteen patients have been reported so far with Alazami syndrome: in actual fact 23 patients from 11 families have been reported [Ivanovski et al, J Hum Genet, 2020]. In the paper in question, which should be cited in the references, the authors can find an overview of the literature in Table 1 and a figure (Figure 4) with all the variants hitherto reported. Therefore the authors should update figure 3 with all the variants so far described.
The authors also highlight the fact that their patient presented acrocyanosis and palmoplantar hyperhidrosis. These clinical features are quite common in the general population and this should be mentioned in the discussion. However, it cannot be excluded that these features might represent unusual phenotypic manifestations of LARP7, or OTOG or a combination of both.
We think that once the above-mentioned modifications have been carried out, the work should most definitely be accepted for publication, but we would like to suggest some minor changes:
-Page 2 Line 44: The authors should write that 23 patients have been reported
-Page 2 Line 59: The authors should write "This study reinforces..." instead of ...reinforce....".
-Page 2 Line 60: The authors should write "..in the presence of.....".
-Page 2 Line 64: The authors should write "This study was in accordance with........".
-Page 2 Line 77: The authors should write "...deleted regions were....".
-Page 3 Line 127: The authors should write "The patient is an 11..." instead of "...a 11........".
-Page 3 Line 127: The authors should write "...first..." instead of "fist...."
-Page 3 Line 132: The authors should write "He never attained...".
-Page 4 Line 155:The authors should write "The finding was made in accordance with.......".
-Page 5 Line 161: The authors should write "Variants' mapping..." or "The mapping of variants...."
Author Response
To Editor-in-Chief
Genes: Special Issue "Selected Papers from the International Workshop on Fragile X and Other Neurodevelopmental Disorders"
March 16, 2020
Subject: genes-741715 revised version submission
Dear Editor,
Thank you very much for giving us an opportunity to submit the revised manuscript to your esteemed journal. In agreement with you, we did all the corrections suggested. The following are Point-by-point response to the Reviewer 2 criticism. We hope that these changes meet with your approval.
Author's Reply to the Review Report (Reviewer 2)
The paper is thorough, interesting and well written.
The authors describe a case with novel deleterious variants in LARP7 and OTOG. This case underlines the importance of combined genomic approaches to solve complex phenotypes and the role of parental consanguinity in prioritizing candidate genes mapping in regions of homozygosity (ROHs). They point to the importance of SNP-Array in identifying these regions initially. This paper expands the molecular spectrum of the genes LARP7 and OTOG and confirm the associated phenotypes.
However, in order to improve the knowledge of these phenotypes it would be very useful to have the photographs of the patient. Furthermore, it would be helpful if the authors indicated the patient’s nationality. ANSWER: We agree with this reviewer concerning the utility to show the facial features of this patient in a picture. Unfortunately, the family, though fully approving the publication of the clinical and molecular data (see attached consent form), did not agree to show the uncovered face of the patient. Regarding the patient’s nationality, we introduce this information on the first line of the section 3.1. Clinical Description: “The patient is a 11-year-old male, son of fist cousins, unaffected parents Caucasian origin (Southern Italy)”
The authors, moreover, state that nineteen patients have been reported so far with Alazami syndrome: in actual fact 23 patients from 11 families have been reported [Ivanovski et al, J Hum Genet, 2020]. In the paper in question, which should be cited in the references, the authors can find an overview of the literature in Table 1 and a figure (Figure 4) with all the variants hitherto reported. Therefore the authors should update figure 3 with all the variants so far described. ANSWER: As suggested, we considered the paper published recently by Ivanovski et al [J Hum Genet, 2020] for review of the literature. Following this indication, we updated the figure 3 and the main text.
In detail, in the section “Discussion”, line 216-225, we have the new review of the literature reported as follow:
“Currently, 24 patients from 12 families have been described with Alazami syndrome and recessive variants in LARP7 [5]. Of the 14 identified variants, 12 were frameshift, 1 non-sense, and 1 was predicted to cause abnormal splicing. Therefore, all deleterious variants causing Alazami syndrome are presumably null alleles [5]. Our findings are in line with the literature, as the variant detected in our patient is a frameshift predicted to lead to the deletion of the last 217 amino acids and, consequently, the RNA recognition motif 2 (RRM2), presumably leading to the formation of a truncated protein [p.(Arg366Thrfs*2)] or nonsense-mediated mRNA decay. This previously unpublished variant is located near to the variants reported in the original report as well as that described by Hollink and colleagues [19], and bring to 15 the number of identified deleterious variants in LARP7 (Figure 3).”
Also, in the reference section the fifth article reported (L.Wojcik, M.H.; Linnea, K.; Stoler, J.M.; Rappaport, L.; Updating the neurodevelopmental profile of Alazami syndrome: Illustrating the role of developmental assessment in rare genetic disorders. Am J Med Genet A. 2019, 179, 1565-1569) has been replaced with the new one: “5. Ivanovski, I., Caraffi, S.G., Magnani, E., Rosato, S., Pollazzon, M., Matalonga, L., Piana, S., Nicoli, D., Baldo, C., Bernasconi, S., Frasoldati, A., Zuffardi, O., Garavelli, Alazami syndrome: the first case of papillary thyroid carcinoma. J Hum Genet. 2020, 65, 133-141”.
The authors also highlight the fact that their patient presented acrocyanosis and palmoplantar hyperhidrosis. These clinical features are quite common in the general population and this should be mentioned in the discussion. However, it cannot be excluded that these features might represent unusual phenotypic manifestations of LARP7, or OTOG or a combination of both. ANSWER: In agreement with the reviewer, we add the sentence (page 9, line 240-244) “The concurrence of these features might be causal, as both are unspecific and are common in the general population. Nevertheless, at the moment, we cannot exclude that acrocyanosis and palmoplantar hyperhidrosis might represent unusual phenotypic manifestations of LARP7, or OTOG or a combination of both.”
We think that once the above-mentioned modifications have been carried out, the work should most definitely be accepted for publication, but we would like to suggest some minor changes:
-Page 2 Line 44: The authors should write that 23 patients have been reported ANSWER: For the review of the literature we considered the paper published recently by Ivanovski et al [J Hum Genet, 2020] and updated the data reported.
-Page 2 Line 59: The authors should write "This study reinforces..." instead of ...reinforce....". ANSWER: We changed the text as suggested
-Page 2 Line 60: The authors should write "..in the presence of.....". ANSWER: We changed the text as suggested
-Page 2 Line 64: The authors should write "This study was in accordance with........". ANSWER: We changed the text as suggested
-Page 2 Line 77: The authors should write "...deleted regions were....". ANSWER: We corrected the text as suggested
-Page 3 Line 127: The authors should write "The patient is an 11..." instead of "...a 11........". ANSWER: We changed the text as suggested
-Page 3 Line 127: The authors should write "...first..." instead of "fist...." ANSWER: We corrected the text as suggested
-Page 3 Line 132: The authors should write "He never attained...". ANSWER: We changed the text as suggested
-Page 4 Line 155:The authors should write "The finding was made in accordance with.......". ANSWER: We changed the text as suggested
-Page 5 Line 161: The authors should write "Variants' mapping..." or "The mapping of variants...." ANSWER: We changed the text as suggested

Round 2
Reviewer 1 Report
The authors have revised the manuscript. For a number of my comments this was approriate. However, for a number of comments this was not sufficient.
The most important point is concerns my question on whether other variants in genes known to be associated with hearing impairment might have contributed to the hearing impairment. The hearing impairment in the patient is more severe than expected for defects in OTOG. The authors stated that there were no additional variants according to the ACMG guidelines. However, the authors applied the allele frequency of 0.002 for recessive hearing impairment. This is too stringent for hearing impairment and especially for the very common c.35delG GJB2 variant and several other variants in GJB2. THe authors should check all variants in this gene let's say with a frequency <10%. For other genes associated with recessive hearing impairment they should use <= 0.5%.
The authors should critically their manuscirpt or ask someone else to do that. I indicated in my comments that there are many mistakes/ misspellings and I gave a few examples. The authors only corrected the examples and did not go through the complete mansucript. Again 2 examples!!!: line 130: 'of ' is missing between 'parents' and 'Causcasian'. Also, in the legend of Fig 3: enconded should be encoded. Lines 219-223: this sentence should be rephrased into a single or 2 correct ones. The authors should check and correct the changes they made in the references.
Author Response
Authors Reply to the Review Report
The authors have revised the manuscript. For a number of my comments this was approriate. However, for a number of comments this was not sufficient.
The most important point is concerns my question on whether other variants in genes known to be associated with hearing impairment might have contributed to the hearing impairment. The hearing impairment in the patient is more severe than expected for defects in OTOG. The authors stated that there were no additional variants according to the ACMG guidelines. However, the authors applied the allele frequency of 0.002 for recessive hearing impairment. This is too stringent for hearing impairment and especially for the very common c.35delG GJB2 variant and several other variants in GJB2. The authors should check all variants in this gene let's say with a frequency <10%. For other genes associated with recessive hearing impairment they should use <= 0.5%.
ANSWER: We sincerely apologize for this lack of information. Indeed, the exome sequencing raw data were accurately scrutinized for additional molecular defects since before the first submission. At this time, we can confirm the absence of clinically relevant variants in all genes associated with Mendelian/digenic/oligogenic hereditary deafness also including GJB2 by applying the cut-offs suggested by the reviewer. In order to clarify this point in the manuscript, we added the following (lines: 191-197): “No further variant classified as pathogenic or likely pathogenic according to ACMG guidelines in other genes, previously associated with phenotypes compatible with the clinical features reported by the patients, were identified by the bioinformatics analysis. In particular, we did not find any additional deleterious variants in other genes responsible for hereditary hearing loss, and annotated in Hereditary hearing Loss Database (https://hereditaryhearingloss.org/), or the other satellite symptoms, including palmoplantar hyperhidrosis and acrocyanosis.”
In addition, in the section 2.3 Whole exome sequencing (WES), lines 119-121, we added the cut-off suggested: (vii) variants in GJB2 gene setting up as cut off a MAF <= 0.1 and variants for other genes associated with recessive hearing impairment and reported in Hereditary hearing Loss Database (https://hereditaryhearingloss.org/) setting up as cut off a MAF <= 0.5.
The authors should critically their manuscirpt or ask someone else to do that. I indicated in my comments that there are many mistakes/ misspellings and I gave a few examples. The authors only corrected the examples and did not go through the complete mansucript. Again 2 examples!!!: line 130: 'of ' is missing between 'parents' and 'Causcasian'. Also, in the legend of Fig 3: enconded should be encoded. Lines 219-223: this sentence should be rephrased into a single or 2 correct ones. The authors should check and correct the changes they made in the references.
ANSWER: We revised these two examples, and rephrased the indicated sentence (lines 225-229 as follows: Our findings are in line with the current literature, as the variant detected in our patient is a frameshift predicted to cause the deletion of the last 217 amino acids, and therefore the loss of the RNA recognition motif 2 (RRM2). At functional level, this event likely lead to the formation of a truncated protein [p.(Arg366Thrfs*2)] or nonsense-mediated mRNA decay.) and corrected the changes made in the references.
In addition, as suggested, the complete mansucript was reviewed by a native speaker who made corrections.
Thank you for your attention.
Yours sincerely,
Dr. Orazio Palumbo, PhD
Medical Genetics Specialist
Fondazione IRCCS Casa Sollievo della Sofferenza
Division of Medical Genetics
71013 San Giovanni Rotondo (FG)
Italy
Tel. +39 0882 416350
Fax. +39 0882 411616
Email: o.palumbo@operapadrepio.it
